# How did the beginnings of the global COVID-19 pandemic affect mental well-being?

**Julie Levacher** [ID] *[☯], **Frank M. Spinath**, **Nicolas Becker**[¤], **Elisabeth Hahn**[☯]

Department of Individual Differences & Psychodiagnostics, Saarland University, Saarbrucken, Germany

☯ These authors contributed equally to this work.
¤ Current address: Department of Individual Differences & Psychodiagnostics, Greifswald University, Greifswald, Germany
* julie.levacher@uni-saarland.de

**Data Availability Statement:** Pre-registration is available on OSF at https://osf.io/8db7p/. The data cannot be shared publicly because the consent of the participants only includes the analysis for research questions and not the digital provision of

## Abstract

The present study aims to investigate longitudinal changes in mental well-being as well as the role of individual differences in personality traits (Big Five) and the level of Personality Organisation during the first lockdown of the COVID-19 pandemic in Germany. Overall, 272 adults ($M_{age}$= 36.94, $SD_{age}$= 16.46; 68.62% female, 23.45% male, 0.69% non-binary) took part in our study with four weekly surveys during the lockdown as well as a follow-up one month after restrictions were lifted. To analyse the development of mental well-being during and shortly after the first lockdown in Germany latent growth curve models (LGCM) were calculated. The considered facets of well-being differ by their trajectory. Additionally, results suggest that the lockdown did not affect all facets to the same extent. While Life Satisfaction decreases in the short term as a reaction to the lockdown, Stress and Psychological Strain were reduced after the second week of contact restrictions. When adding personality characteristics, our results showed that Neuroticism and Conscientiousness were the two dimensions associated most strongly with SWB during the first month of the pandemic. Thus, our research suggests that personality traits should be considered when analysing mental well-being.

## Introduction

### Background

In the first months of 2020, a crisis shocked the world: the novel coronavirus (SARS-CoV-2) spread rapidly around the world, leading to the onset of the COVID-19 pandemic in only a few months. The resulting COVID-19 pandemic forced many countries – including Germany, the country under study – to impose strict restrictions to prevent further outbreaks that could possibly lead to the collapse of the health care system. This led to a complex pattern of consequences not only for those infected by the virus, but also for 'healthy' individuals and society as a whole [1, 2]. In addition to a variety of health problems caused directly by COVID-19 or indirectly by inability to access health care and associated challenges, people had and still have to deal with enormous social and economic challenges. Isolation resulting from social

the data. All our analyses are based on the use of sensitive data. For our findings it is therefore mandatory to have on the one hand the information concerning mental health and well-being and on the other hand those including age, sex, high-risk group belonging and status as an essential worker. Therefore, dis-identifying leads to an unusable dataset. We recommend contacting the secretary of the Ethic Committee of the Faculty of Human and Business Sciences of Saarland University (d. mast@mx.uni-saarland.de) for data access.

**Funding:** The authors received no specific funding for this work.

**Competing interests:** The authors have declared that no competing interests exist.

distancing and the closure of schools and workplaces likely affects almost all individuals and increases the experience of stress, anxiety, fear, and loneliness. But to what extent do individuals differ in their responses to such restrictions based on their personalities and coping mechanisms?

From end of March 2020, when this study was conducted, into mid-2021, the COVID-19 pandemic led to a plethora of restrictions on everyday life. In Germany, these restrictions included the closure of day care centres, schools, and universities, many shops, restaurants and bars, and other workplaces between March and May 2020. Other challenges during this time included loss of or reduced income, isolation from others, fear for one's own health and for the safety of loved ones [3]. Finally, strict lockdown measures such as "stay at home orders" restricted the freedom of every individual in society. These constraints may have led to a variety of fear responses including health and economic fears, which can mutually affect one another. It is possible that fear of the future or existential fears, which might arise through situations such as short-time work, are also related to fears of contagion, which are reinforced by social isolation [4]. This example shows that economic, health-related and social fears are very different, but can be mutually dependent and thus lead to an overall fear. At the same time, it is possible that economic fears may be more influenced by individuals' occupations and living conditions than health fears. In addition, psychological resources play a role in the expression of anxiety [4]. In this respect, the restrictions imposed in response to the COVID-19 pandemic are experienced very differently for each individual and can play a role in how well people cope with the pandemic.

Nevertheless, as almost everyone experienced the consequences of the pandemic, the COVID-19 pandemic can be seen as a global stressful life event [5]. These are generally defined as events that require changes in people's normal activities and habits and often lead to readjustment [6]. However, healthy readjustment also requires individual resources (such as cognitive skills and mental health) that not all people possess to the same extent. Therefore, it is likely that not all people will be equally successful in this readjustment process. The aim of this study was to identify relevant personal characteristics that come into play in coping with such difficult life conditions.

## Well-being during the COVID-19 pandemic

In general - even before the COVID-19 pandemic - substantial research had addressed the importance, determinants, and implications of subjective well-being (SWB) as a core construct for assessing quality of life. SWB can be seen as a wide-ranging concept, including general Life Satisfaction and various forms of domain-specific satisfaction as its cognitive components (e.g., Life Satisfaction), and positive (e.g., Joy, Pleasure) as well as negative affect (Anxiety, Stress, Loneliness) as its emotional components [7, 8]. Thereby, a substantial body of research has focused on the extent to which major life events can influence individual SWB. For example, marriage, divorce and widowhood are events that can affect SWB, at least temporarily. Earlier research in this area demonstrated, however, that after initial increases or decreases depending on the nature of the event, individual well-being tended to return to the initial level [9–11]. This high consistency of SWB in empirical studies supports the set-point theory, which states that adults' levels of SWB vary around a stable individual set-point; thus, even dramatic negative or positive events can usually only impact SWB for a certain period of time [10]. In contrast, some studies such as one by Hahn and Colleagues [12] discovered patterns of Life Satisfaction contrary to set-point theory. In investigating unemployment, they found that even after finding a new job, not all individuals returned to their original level of Life Satisfaction. These differences between individuals were predominantly attributable to differences in

personality traits. In sum, set-point theory may not be valid for all individuals and every life event [13, 14], leading to the question of to what extent a pandemic influences Life Satisfaction and how different personalities may react to it.

First, it could be assumed that a dramatic event such as the COVID-19 pandemic should initially influence the trajectory of SWB in a negative way. This was supported in a study by Kuper and colleagues [15], who found a reduction in SWB for a German sample during the first lockdown at the end of March 2020, while stringent restrictions were in place, in comparison to a retrospective rating of SWB before the pandemic as well as a follow-up in October 2020. Similar results showing an initial decrease in well-being and an increase in mental health problems were found in a longitudinal analysis of people living in the UK [16] as well as for an Australian sample at the beginning of the second wave of infections (July/August 2020) [17]. In contrast, other studies came to contradictory conclusions concerning changes in well-being during the COVID-19 pandemic. Investigating a UK sample longitudinally, O'Connor and colleagues [18] reported an overall increase in well-being. Increasing well-being and health scores were also reported by Recchi and colleagues [19], who compared French panel data before and after the outbreak of COVID-19. These surprising and somewhat counterintuitive findings indicate that a significant number of individuals, particularly those who are not directly affected by the virus, evaluate the situation in a positive manner, resulting in a rise in well-being scores. However, Kiwi and colleagues [20] analysed Swedish data and found that well-being in 2020 remained relatively stable compared to previous years, which is in line to findings from a Chinese study [21] reporting no longitudinal changes in stress, anxiety, or depression scores. A similar conclusion was drawn by Zacher and Rudolph [22] for a German sample. This longitudinal study likewise found no significant changes in SWB between December 2019 and the early phase of the pandemic in March 2020. In summary, the results on well-being are ambiguous and different studies have come to contradictory conclusions.

One possible reason for the contrasting results may be that the studies cover different countries. Due to different levels of infection, countries differed greatly in the restrictions they imposed and for what time periods. However, there are some commonalities in the results. Thus, an initial decrease in well-being was shown in Australia and Germany during lockdown periods [15, 17]. A second possible reason for contradictory findings could be the large number of different indicators falling under the umbrella of SWB. Due to the great variety of SWB facets, it is advisable to always capture the major components of well-being (cognitive components, positive and negative affect) [7]. Often, however, only facets of seemingly higher relevance are included in a given study. For example, while Anglim and Horwood [17] recorded SWB by measuring Life Satisfaction, positive and negative affect with different scales, Recchi and colleagues [19] asked participants to rate how often they felt 'nervous', 'low', 'relaxed', 'sad', 'happy' and 'lonely', without focussing on Life Satisfaction. Additionally, studies have differed in their frequency of data collection. While Recchi and colleagues [19] compared their measured well-being levels in 2017-2019 with three time points during the lockdown for one population, Anglim and Horwood [17] draw a comparison between a pre-COVID sample from 2017 and a COVID sample with one data collection period during a lockdown period. This is another reason why comparing different well-being scores from different studies may be problematic.

Previous studies have come to different conclusions about the relationship between SWB and the COVID-19 pandemic (negative, neutral, positive). In addition to the survey time points, prior studies also differ in their samples' socio-demographic distribution (e.g., age, occupational status) and in terms of how affected sample participants were by the COVID-19 pandemic. Thus, differences in well-being associated with the pandemic could be explained by various moderators that could influence both well-being and its course. However, it is also

conceivable that reactions and consequences can vary greatly from person to person. Therefore, it is necessary to consider both an inter- as well as an intraindividual perspective. Here, the translational stress theory by Lazarus could be applied, which described consequences of stressful life events on health [23]. In this model, both characteristics of the life event itself and the abundance and accessibility of resources of an individual and societal nature are crucial antecedents of appraisal outcomes. Their connection with health, which also includes well-being, is mediated by appraisal and coping processes. With respect to the COVID-19 pandemic, the model provides us with a framework for examining relevant factors associated with the psychological impact of the COVID-19 pandemic and associated restrictions.

To capture how individuals react to the global pandemic and the respective restrictions dictated by the government immediately and in the long run, a longitudinal assessment of mental well-being, including SWB means and other different influencing aspects of well-being is mandatory. Thereby, it makes sense to capture both cognitive and emotional aspects of SWB, as a comparative process is involved in cognitive aspects. First, Life Satisfaction as the main cognitive factor of SWB is an important indicator, since it involves evaluating one's actual and target state in comparison to others, previous states and ideal states [24]. Life Satisfaction has shown to be relatively stable but can be influenced by major life events [25, 26]. At this point, the COVID-19 pandemic can be seen as a collective major life event. Since the COVID-19 pandemic is a global event that affects everyone, this comparison process may not be so negative and Life Satisfaction may not drop so much. Only in the course of time does it become apparent in numerous individual comparisons, perhaps also with one's own ideal and earlier stages, that the event is judged negatively, and Life Satisfaction therefore does not return to the starting point. This would not be the case with emotional aspects, which are closely tied to SWB [27, 28]. With these, one would expect stronger reactions, even if only in the short term. For this reason components of mental health should also be considered. Here, two facets could be especially relevant. Loneliness, which is regardless of age associated with quality of social relationships [29] is most likely directly influenced by contact restrictions. The other interesting facet is Stress, which is closely related to quality of life [30] and which may have been induced by a variety of aspects related to the pandemic, e.g. the fact that it's a novel situation that required some sort of adaptation or in particular that one has to deal with closed schools, home teaching or the need to work from home. This highlights that the three aspects, namely Life Satisfaction, Loneliness and Stress are likely to be affected differently by the pandemic and that, on the person's individual side, different resources, external circumstances, and personality traits also influence how these mental well-being components change as a result of the pandemic.

The perspective on interindividual differences in reaction to the same event is supported by a recent study that examined the complexity of psychological responses to quarantine [5]. Many different stressors on an objective but also individual level were identified, including the duration of quarantine, fear of infection and health risks, boredom, and frustration, all of which could influence the psychological impact of quarantine. Therefore, in addition to considering more general effects of the pandemic, an individual differences perspective is needed that focuses on variation in the causes and consequences of the longitudinal course of well-being in response to the COVID-19 pandemic.

## The influence of personality on the expression of well-being

It is widely known that personality traits are strongly linked to well-being, while socio-demographic factors (e.g., age, gender, income) seem to be only moderately related [23, 31]. In particular, the Big Five personality traits have been found to influence well-being in consistent

ways. Neuroticism, for example, has a moderately negative relationship with Life Satisfaction. In contrast, Extraversion and Conscientiousness have moderately positive relationships with Life Satisfaction. Agreeableness appears to be only slightly positively related to Life Satisfaction [13, 14]. In addition, Agreeableness and Conscientiousness have been postulated to be prerequisites for the creation of desirable living conditions and is thus associated with higher psychological well-being [32]. A study by Steel and colleagues [14] showed that personality explains up to 39% of the variance in quality-of-life measures.

Furthermore, pre-event personality characteristics can have a decisive impact on how a situation is perceived, which coping mechanisms are applied, and the extent of personal suffering. Since personality structure is thought to have influenced coping and well-being during the COVID-19 pandemic, its influence has been considered in several recent studies.

Extraversion, and to a lesser extent Agreeableness, seemed to be protective. A possible reason for this is that people who scored high on these two dimensions were able to reduce negative affect during isolation by activating coping mechanisms [33]. In contrast, no difference between extraverted and introverted participants regarding their level of Loneliness was found [34]. Moreover, higher scores in the personality dimensions of Extraversion and Neuroticism were associated with a more negative appraisal of the pandemic [35]. In particular, the personality dimension Neuroticism seems to influence how people dealt with the pandemic. Thus, individuals with high Neuroticism scores seemed more concerned and worried about COVID-19 [36, 37]. This was particularly evident by the fact that these participants paid more attention to information about the COVID-19 pandemic and experienced more negative affect in dealing with the pandemic. It was therefore expected that participants with high scores on Neuroticism would experience a stronger decline in SWB scores than the rest of the sample.

Overall, the two personality dimensions Extraversion and Neuroticism seemed to be most strongly related to well-being during the COVID-19 pandemic. However, the correlations between well-being and personality have primarily been shown in cross-sectional studies [33, 35], which present 'snapshots' analysed with respect to group differences. However, the effects might be of different strengths depending on the time and duration of lockdown and should therefore be considered more closely.

The established correlations (positive correlation with well-being: Extraversion, Agreeableness, Conscientiousness and Openness; negative: Neuroticism) also persist during the pandemic [35]. The opposite pattern of correlations was shown with regard to anxiety and depressive symptoms [33]. Concerning negative appraisal, the expected effect was found, namely that people scoring high on Neuroticism react more negatively to an event like the COVID-19 pandemic. For Extraversion, however, the results are unexpected. Normally, extroverts react less negatively to a negative event. In the case of the pandemic, however, they also react strongly negatively, presumably because of the contact restrictions during the lockdown. This could be because being social is normally a protective factor, but it cannot take effect during the lockdown. It is therefore conceivable that during the pandemic, normally protective or hindering resources and characteristics have a different impact on mental well-being due to the numerous restrictions in life, and that interactions occur. This suggests that although the pandemic is a major event, the influence is not so strong that it obscures variation. Thus, inter-individual differences can still be seen.

## The influence of psychological functioning on the expression of mental well-being

However, other personality dimensions besides the Big Five might also be related to mental well-being and potentially be relevant to the question how people deal with the pandemic.

Indeed, personality can also be understood in a clinical sense at a more basal level of psychological functioning. According to Kernberg's (1984) model of Personality Organisation a person's level of Personality Organisation is related to the occurrence, severity and course of mental disorders. This model assumes that the degree of Personality Organisation is defined by the effectiveness or impairment of basic adaptive abilities, including one's own functioning (e.g., self and other perception, self-esteem regulation) and interpersonal functioning (e.g., regulating desires, feelings, and behaviour according to the social situation). The differentiated view of personality is particularly useful because it has been shown that Personality Organisation and the Big Five personality traits are distinct but interrelated constructs that both contribute to mental health [38].

It therefore has a decisive influence on how healthy people react to stress and how stressed they feel. As the COVID-19 pandemic can be seen as a challenge for mental health [4], Personality Organisation should also be considered in relation to COVID-19.

### The influence of socio-demographic factors on the expression of well-being

However, individual differences in well-being during the COVID-19 pandemic cannot only be attributed to personality. In order to identify further influencing factors, different group comparisons have been carried out. Accordingly, people with health-related risk factors (chronic illness, pregnancy, old age, disability) as well as employed people (those working from home as well as those going into the workplace) seemed to be more stressed by the pandemic [19]. Further influencing factors were found by Möhring and colleagues [39]. In their study with German participants, a decrease in job satisfaction was found among people who had to work reduced hours and among mothers. In contrast, the authors found that fathers' well-being was less negatively affected in comparison to mothers. In a UK sample, O'Connor and collegues [18] showed that women in particular and young adults aged 18-29 had the greatest mental health problems. These results are consistent with Daly and colleagues [16], who found the worst mental health outcomes in their UK sample for 18- to 34-year-old participants. However, in terms of socio-economic status, this study showed that participants with higher income and education also had more mental health problems. This is in direct contrast to other studies finding that higher socio-economic status was associated with better mental well-being [18, 40]. Overall, the literature is inconsistent regarding the influence of socio-economic status, which may be due to different operationalisations and study designs in different countries. This assumption could be confirmed by Recchi and colleagues [19], who found an increase in well-being with the onset of lockdown independent of monthly income among their French subjects. In contrast, group differences emerged regarding place of residence and occupation. People who spent many hours working from home had a higher Stress level, while those who lived in a big city (Paris) had lower well-being values than the rest.

To summarize, other possible influencing factors besides personality have already been studied, albeit with mixed results. These include socio-economic status, age, and gender. In addition, occupation and place of residence also seem to have an influence on well-being.

### Aims of the present study

While previous studies have mostly conducted group comparisons in cross-sectional designs, the present study aims to investigate longitudinal changes in mental well-being as well as the role of individual differences in personality traits and the level of Personality Organisation in these longitudinal trajectories of mental well-being during the early stages of the COVID-19 pandemic. For this purpose, four different mental well-being measures (Life Satisfaction, Stress, Psychological Strain, and Loneliness) were captured longitudinally during the first

months of the COVID-19 pandemic in Germany. The survey period includes the first lock-down and a follow up, after the restrictions were lifted, so that the influence of this lockdown period on mental well-being can also be considered. In contrast to other studies, we placed a strong emphasis on the analysis of the trajectory of mental well-being by administering questionnaires at short intervals of weekly surveys during the lockdown. This approach allows for a closer inspection of changes throughout this phase of the pandemic. In addition, analyses are based on a diverse sample that showed a high participation rate across measurement points. This makes it possible to additionally consider individual differences in the course of mental well-being during the first months of the pandemic. In doing so, we not only rely on the commonly used five-factor model of personality [41] but also on Kernberg's (1984) model of Personality Organisation. To summarize, the following questions have been addressed:

1. Did mental well-being change over the first months of the COVID-19 pandemic, including the first lockdown?

2. Did individual differences in personality characteristics (Big Five and Personality Organisation) influence changes in mental well-being over the first months of the COVID-19 pandemic?

3. Are there any group differences (essential workers, high-risk population) with regard to changes in mental well-being over the first months of the COVID-19 pandemic?

## Methods

### Study design, participants and procedure

On March 16, 2020, the German government announced a nationwide lockdown beginning on March 22, 2020 [42]. At the time, this represented an unprecedented intervention to curb the spread of COVID-19. Shortly after the release of this statement, a link to register for our study, termed CoCo ("Coping with Corona"), was shared through various social media channels as well as via email to the companies of essential workers in the Saarland region. On the study website, participants were informed that their data would be processed anonymously, that they could end their participation at any time without giving a reason, and that they needed to be a minimum of 18 years old to participate. In accordance with the original study design, three further surveys at weekly intervals (April 2, 2020; April 9, 2020; April 16, 2020) and a final survey four weeks after the fourth survey (May 14, 2020) were conducted. This design allowed us to capture an important period of time starting with the announcement of a national lockdown and ending after the first lockdown was lifted. The survey waves in relation to the number of new infections in Germany and state of mitigation measures can be seen in Fig 1.

Overall, 290 adults registered for the study, of whom 272 took part in the first survey wave [$M$ = 36.94 years ($SD$ = 16.46 years, $19 \leq$ age $\leq 80$), 68.62% female, 23.45% male, 0.69% non-binary]. A total of 263 participants took part in the second wave (96.7% of participants in the first wave), which corresponds to an excellent response rate. This was followed by a slight decline in the third wave, with 250 participants (91.9% of the first wave), 243 (89.3% of the first wave) in the fourth wave, and 220 (80.9% of the first wave) in the fifth and final wave survey. 199 people (73.2% of the participants) completed all five waves. We managed to capture a broad and diverse sample that was not restricted to students or another sub-population, as is often the case in academic research. Further socio-demographic data supporting this assumption can be found in Table 1. Prior to any analyses, incomplete responders ($N$ = 91) were compared to those who participated in all survey waves ($N$ = 199). Overall, no significant

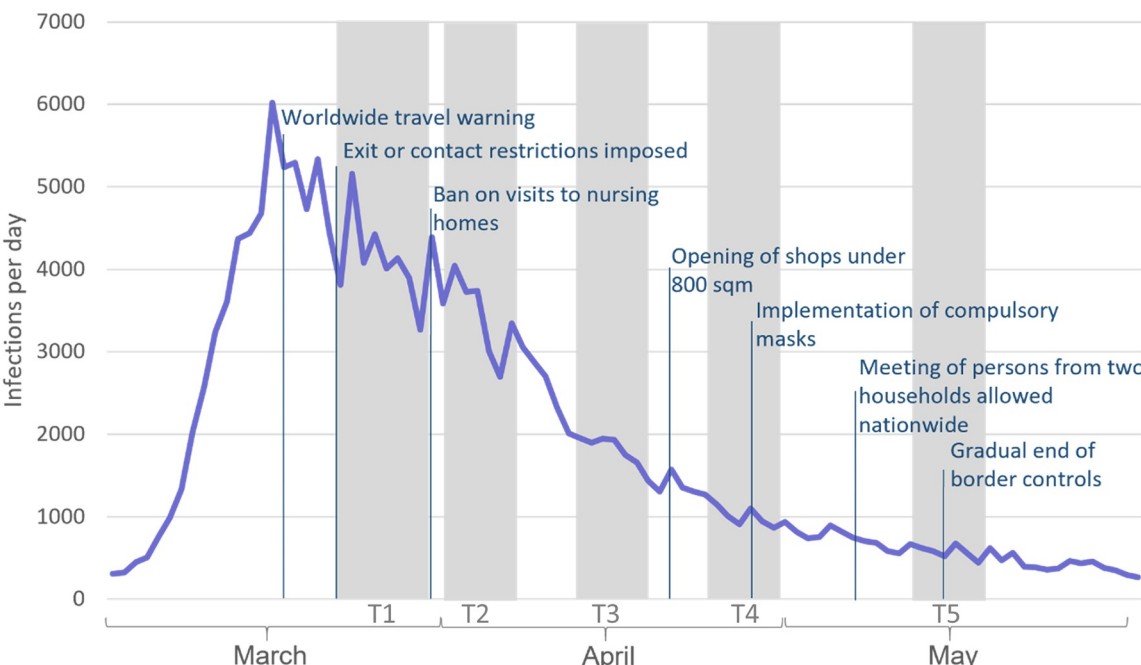

**Fig 1. Survey periods in connection with the number of new infections and key points of the regulatory orders in Germany.**
Survey periods are shown in grey, the number of new infections in purple.

differences between the two groups could be found. They did not differ in terms of age ($M_{complete}$ = 37.55, $SD_{complete}$ = 16.74; $M_{incomplete}$ = 35.1, $SD_{incomplete}$ = 15.63; $t(268)$ = -1.09, $p$ = .279, $d$ =.138) or gender ($\chi^2(2)$ = 2.35, $p$ = .273). A summary of further comparisons can be found in S1 Table. These results show that there was no systematic dropout. Therefore, the use of the full data set for subsequent statistical analyses is justified. To handle missing data, the full information maximum likelihood (FIML) method was used [43]. The advantage of the FIML method is the replacement of missing values when determining standard errors for the estimated parameters, which makes it possible to accurately assess uncertainty.

## Measures

**Mental well-being.**  The construct mental well-being was measured using different indicators. A positive dimension was assessed in terms of *Life Satisfaction*, as main cognitive factor of SWB. In addition, *Stress*, *Loneliness* and *Psychological Strain* were assessed to capture mental well-being. Importantly, these two dimensions are partially independent indicators and do not represent two ends of a common continuum [7, 8].

**Life satisfaction.**  Life Satisfaction was measured longitudinally using the short scale Life Satisfaction-1 (L-1; [44]), a single-item scale to measure general Life Satisfaction. Prior analyses have shown that this scale has a high positive correlation with a multiple item scale ($r$ = .74), indicating good convergent validity [44]. Retest reliability after six weeks (rtt = .67) was also acceptable. For this study, the response format was adapted. In contrast to the original 10-point rating scale, our study participants were able to indicate their Life Satisfaction using a slider from "completely dissatisfied" (0) to "completely satisfied" (100). At each wave, participants rated their current Life Satisfaction ("All things considered, how satisfied are you with your life these days?"). Because we wanted to be able to capture even small differences between the five waves, the adapted response format was necessary.

**Table 1. Socio-demographic information of the sample.**

| | Overall (N = 290) | Complete (N = 199) | Incomplete (N = 91) |
|---|---|---|---|
| **Age** (years) | | | |
| M (SD) | 36.94 (16.46) | 37.56 (16.74) | 35.1 (15.63) |
| Mdn [minimum; maximum] | 29 [19; 80] | 29.5 [19; 77] | 28 [19; 80] |
| **Gender** | | | |
| Female | 199 (68.62) | 151 (75.88) | 48 (52.75) |
| Male | 68 (23.45) | 46 (23.12) | 22 (24.18) |
| Diverse | 2 (0.69) | 1 (0.5) | 1 (1.1) |
| Missing | 21 (7.24) | 1 (0.5) | 20 (21.98) |
| **Education** | | | |
| Elementary and secondary school leaving certificate | 2 (0.69) | 1 (0.5) | 1 (1.1) |
| Mittlere Reife. Realschulabschluss or equivalent qualification (comparable with intermediate or general secondary school leaving certificate) | 9 (3.1) | 6 (3.02) | 3 (3.3) |
| Completed apprenticeship | 14 (4.83) | 10 (5.03) | 4 (4.4) |
| Vocational baccalaureate. entrance qualification for a university of applied sciences | 19 (6.55) | 14 (7.04) | 5 (5.5) |
| A-levels. university entrance qualification | 119 (41.03) | 86 (43.22) | 33 (36.26) |
| Technical college/university degree | 100 (34.48) | 74 (37.19) | 26 (28.57) |
| Other qualification | 9 (3.1) | 8 (4.02) | 1 (1.1) |
| Missing | 18 (6.21) | 0 | 18 (19.78) |
| **Residence** | | | |
| in the countryside (less than 2.000 inhabitants) | 21 (7.24) | 16 (8.04) | 5 (5.5) |
| in a rural town/community (2.000-5.000 inhabitants) | 48 (16.55) | 33 (16.58) | 15 (16.48) |
| in a small town (5.000-20.000 inhabitants) | 49 (16.9) | 38 (19.10) | 11 (12.09) |
| in a medium-sized city (20.000-100.000 inhabitants) | 73 (25.17) | 53 (26.63) | 20 (21.98) |
| in a large city (more than 100.000 inhabitants) | 81 (27.93) | 59 (29.65) | 22 (24.18) |
| Missing | 18 (6.21) | 0 | 18 (19.78) |
| **Essential workers**[1] | | | |
| yes | 89 (30.69) | 60 (30.15) | 29 (31.87) |
| no | 93 (32.07) | 70 (35.18) | 23 (25.28) |
| Missing | 108 (37.24) | 69 (34.67) | 39 (42.86) |
| **High-risk group**[2] | | | |
| yes | 69 (23.795) | 56 (28.14) | 13 (14.29) |
| no | 200 (68.97) | 142 (71.36) | 58 (63.74) |
| Missing | 21 (7.24) | 1 (0.5) | 20 (21.98) |

*Notes*. The numbers in brackets indicate the corresponding percentage value (%).

[1]People who work in a company that has an infrastructural role in a state. The profession and the person who performs it therefore have a special relevance and importance.

[2]People who are known to be at higher risk of severe disease progression in the case of COVID-19 infection.

The three additional mental well-being measures, namely Stress, Psychological Strain and Loneliness, were assessed in a similar way as Life Satisfaction.

**Stress and psychological strain.** From the first wave (T1-T5) on, participants were asked to indicate their personal level of Stress ("How much stress are you under?") as well as their level of Psychological Strain ("How much do you feel strained?") weekly. Responses were recorded using a slider (0-100) with the poles "not at all" and "totally and completely".

**Loneliness.** To measure Loneliness, we used the single-item scale from the Center for Epidemological Studies Depression Scale (CESD; [45]). From the second survey wave onwards,

participants were asked about their feelings of Loneliness ("I often feel lonely at the moment"). As on the other SWB items, responses were recorded on a slider with poles labelled "strongly disagree" (0) and "strongly agree" (100). Higher scores therefore indicate stronger feelings of Loneliness.

**Big five personality.**   Big Five Personality was assessed twice (in the first and fifth waves) using the short version of the Big Five Inventory (BFI-K; [46]). This questionnaire encompasses a set of 21 items measuring Extraversion, Neuroticism, Agreeableness, Conscientiousness and Openness to experience. All items are rated on a 5-point Likert scale (1 = "strongly disagree" to 5 = "strongly agree"). Since personality can be assessed in about two minutes with the BFI-K, the questionnaire is an economical instrument. Its validation study (44) indicates that the instrument has sufficient psychometric properties given its brevity (internal consistency: $.59 \leq \alpha \leq .86$; retest reliability: $.76 \leq r_{tt} \leq .93$; $M_{Neuroticism} = 2.88$ to $M_{Conscientiousness} = 3.53$). As in the validation study, mean values for each scale were calculated in this study. Consequently, high values indicate a high expression of the personality dimension, whereas low values indicate a low expression. In our study, similarly sufficient psychometric properties and comparable mean values were found (Extraversion: $\alpha = .83$, $M = 3.75$; Neuroticism: $\alpha = .76$, $M = 2.81$; Agreeableness: $\alpha = .65$, $M = 3.35$; Conscientiousness: $\alpha = .69$, $M = 3.86$; Openness to experiences: $\alpha = .64$, $M = 3.98$).

**Personality organisation.**   To measure structural impairment, the 16 items of the Inventory of Personality Organisation (IPO-16; [47]) were also included in the first and fifth survey waves. Structural impairment is measured in the areas of identity diffusion, primitive defence, and lack of reality check. Responses on all 16 items were given on 5-point Likert scale (1 = "never true" to 5 = "always true"). The internal consistencies reported by Zimmermann and Colleagues for the IPO-16 ($.85 \leq \alpha \leq .90$) and the retest reliability after two months ($r_{tt} = .85$) can be considered good to excellent. Furthermore, the self-assessment has good convergent validity as well as good discriminant validity. In this study, we calculated the mean value for each participant on this one-dimensional instrument. An acceptable internal consistency of $\alpha = .79$ was found. The mean value for structural impairment in our study ($M = 1.96$, $SD = 0.46$) corresponds to the cut-off value in the manual ($M_{cut-off} = 1.97$), and individuals with a higher IPO score have a significantly higher probability of developing a personality disorder, whereas lower scores are an indicator of mental health.

**Additional variables.**   Furthermore, we asked all participants to answer some situational questions. These covered the areas of work (e.g., (currently) working from home, consideration as an essential worker), whether they currently had or had recovered from a COVID-19 infection, as well as being in quarantine and restrictions concerning personal contact with others. Additionally, we assessed the participants' educational level, place of residence, and health status (belonging to a high-risk group). Allocation to high-risk groups took place via self-assessment. Participants were referred to the criteria by the Robert Koch Institute (RKI; the leading public health agency in Germany), which classify different groups of people (e.g., elderly people older than age 50/60, people with underlying diseases such as cardiovascular disorders, diabetes, respiratory diseases, liver disease, kidney disease or cancer (regardless of age), immunosuppressed people) as more vulnerable to severe disease following a COVID-19 infection.

## Ethical considerations

The Ethic Committee of the Faculty of Human and Business Sciences of Saarland University confirmed in a written statement that our study meets the standards for the ethics. Thereby, we received the information that we do not need any further formal ethics approval.

To participate in this study and to maximise the transparency, participants had to register on a separate homepage. There they received information concerning the data privacy policy, the anonymity of their shared data as well as the goal of our study. Furthermore, they consented to participate on this homepage, but were free after receiving the link to skip the survey or parts of it. The consent was given in a written type by entering name, e-mail-address and ticking a checkbox. Moreover, participants received the results of our study. Thereby we informed, that we categorised participants for our research questions concerning gender (female, male), their allocation to high-risk groups and whether they were essential workers or not.

## Analytical strategy

The aim of the study was to examine the development of mental well-being during and shortly after the first lockdown in Germany. For this reason, various latent growth curve models (LGCM) were calculated and compared. Each growth curve model considers an intercept growth factor (i) as well as one (or two) slope growth factors (s, q) used to estimate the average rate of change. In total, four different models were considered for each of the four facets of well-being (Life Satisfaction, Stress, Psychological Strain, and Loneliness) in order to depict the trajectory over time as best as possible. It was generally assumed that there would be a decrease in well-being during the first lockdown. In our models, the intercept factor reflects the well-being level after the lockdowns ended, while the slope factor(s) is associated with the trajectory of each facet. It should be mentioned that the intercept was modelled in relation to the last survey point, as our well-being measures were not collected before the lockdown. Based on the set-point theory, we assume that we find a baseline after the lockdown. For each growth factor (i and s/q), means and variances as well as covariances were calculated. While the mean of an intercept can be interpreted as the average of the facet over individuals after the lockdown, the variance of the intercept factor indicates differences between people concerning this final level of well-being. Similarly, the mean of the slope factors indicates the average growth rate across individuals, while the variance reveals differences between individuals concerning this rate. The covariance of the slope and intercept reflects the relationship between the individual intercept and slope values.

**Preparation.** Before analysis, the dataset was checked for normal distribution. Because the mental well-being data were slightly skewed (see Fig 2 and S3 Table), all growth curve models were estimated using maximum likelihood (ML) and bootstrapping [48]. All statistical analyses were performed on the raw data using R version 3.5.1. The GCM were specified using the R package lavaan [49]. We evaluated overall model fit using chi-square statistics ($\chi2$) in combination with the root mean square error of approximation (RMSEA), standardized root mean square residual (SRMR) and the comparative fit index (CFI). For the RMSEA, values below .05 indicate a good fit and values between .05 and .08 are interpreted as acceptable fit [50]. For the CFI, values greater than .90 are interpreted as a good fit, and a SRMR smaller than .08 is an indicator of a good model fit [51]. The standard significance level used was 5% ($p < .05$) and confidence interval level was set to 95%.

**Models.** The aim of Model 1 was to examine impairment during the lockdown. For this purpose, it was assumed that the mental well-being measures were at one level during the lockdown and stabilised up or down back to the base level after the lockdown was lifted. In this model, the slope factor was coded discontinuously as (-)1; (-)1; (-)1; (-)1; 0 (negative values for Life Satisfaction and positive values for Stress, Psychological Strain and Loneliness). For all models, the intercepts (i) of the mental well-being measures were fixed at zero, referring to the time point after the lockdown ended and indicating the average well-being level and its variance across individuals. The discontinuous slope factor (s) refers to the overall decline in

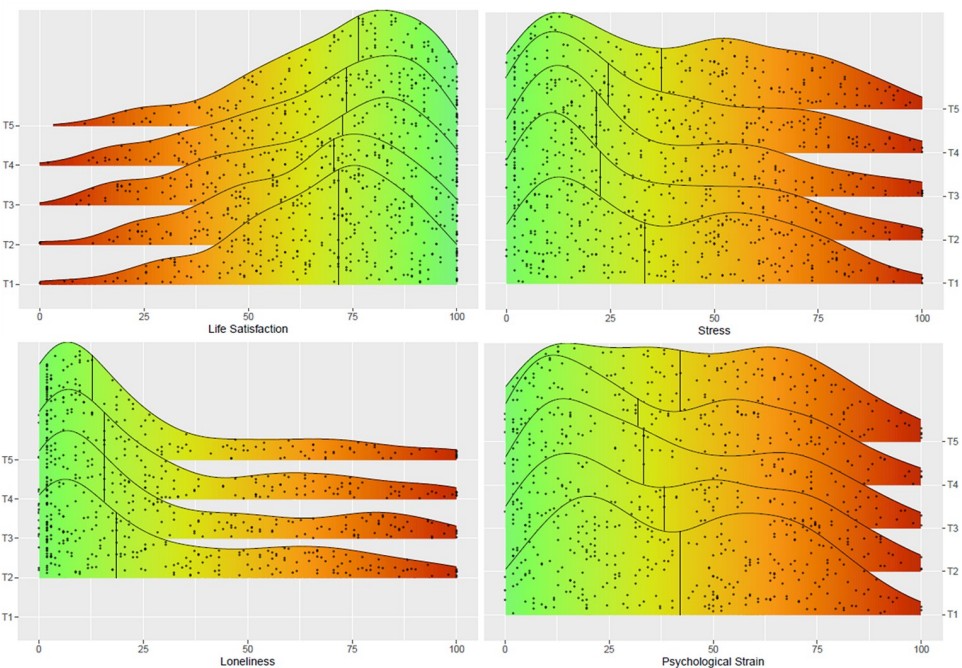

**Fig 2. Distribution of SWB over measurement times.** The black line in each graph represents the median. In terms of Life Satisfaction, higher values mean more Satisfaction (0 – negative vs. 100 - positive), while for Stress, Psychological Strain and Loneliness, high values represent more Stress, Strain and Loneliness (0 – positive vs. 100 - negative).

mental well-being during the lockdown und provides information on the average and variance of the growth rate.

In a second model, a latent growth curve model (LGCM) was specified to investigate whether the lockdown led to a sharp drop in mental well-being, as previous research had indicated that well-being changes more quickly in the period immediately surrounding an event before slowing down over time. This linear trajectory was drawn by specifying the slope parameter with equidistant time scores of (-)4; (-)3; (-)2; (-)1; 0 (negative values for Life Satisfaction). Here too, the intercept reflects the mean level of mental well-being and its variance after the restrictions were lifted, while the linear slope factor (s) tests if there were higher values in mental well-being after the event occurred. Therefore, this growth factor tests a steeper growth rate than the one from Model 1.

Combining Models 1 and 2, Model 3 was set up so that the lockdown vs. no lockdown comparison was represented by Slope 1 (s1) and an additional linear trajectory by Slope 2 (s2). In this model, three random effects were analysed: an intercept factor (i) reflecting the average endpoint after the lockdown, and a discontinuous slope factor (s1) as well as a second linear slope factor (s2) representing the trajectory the well-being measures. The first slope (s1) reflects the overall decline in mental well-being during the lockdown, while the second (s2) expects a large initial effect of the lockdown, which then decreases over time. The variability of all three factors indicates differences between participants in terms of their final well-being and their amount of change. The first slope factor (s1) was coded (-)1; (-)1; (-)1; (-)1; 0 (negative values for Life Satisfaction), while the second slope factor (s2) had a continuous coding (-)4; (-)3; (-)2; (-)1; 0 (negative values for Life Satisfaction).

Furthermore, a non-linear growth change was considered (Model 4). Like in the previous models, the intercept (i) was fixed at zero and refers to the time point after the lockdown ended. To test a non-linear trajectory, a quadratic latent growth parameter was added to Model 2.

Therefore, both the linear and quadratic trends can be tested. Like the linear slope, the quadratic slope also postulates that the pandemic had the strongest effect at the beginning of the lockdown. The contrast can be seen in the trajectory, q (-)16; (-)9; (-)4; (-)1; 0 (negative values for Life Satisfaction). While the linear slope factor (s) tests a linear growth rate, the quadratic growth factor (q) tests a quadratic trajectory, meaning that the initial effect of the lockdown weakens rapidly from week to week. Means and variances are reported for both slope factors (s and q).

To identify the best-fitting model for each mental well-being measure, nested models were compared using chi-squared difference tests. Non-nested models were compared using the Akaike information criterion (AIC).

**Influence of personality.** To examine whether individual differences in personality characteristics (Big Five and Personality Organisation) influence changes in mental well-being over the first months of the COVID-19 pandemic in Germany, the intercept and slope factors were treated as outcomes. To predict differences in the well-being level between participants and patterns of change, the best fitting model from Hypothesis 1 was specified by adding T1 personality factors as covariates. To analyse the combined effect of personality, all personality factors (Big Five and Personality Organisation) were modelled simultaneously, while controlling for age. This approach allows us to estimate the effect of each personality factor while controlling for the effect of all other factors. For all personality factors, influence on final mental well-being level (intercept) and on changes in mental well-being (slopes) were modelled.

**Group differences.** A similar approach was adopted to address group differences (essential workers, high-risk group, gender) with regard to changes in mental well-being over the first months of the COVID-19 pandemic in Germany (Hypothesis 3). Here, essential workers, gender (after dropping non-binary persons), and high-risk group status were dummy coded and used in a regression as independent variables to predict group differences in the trajectory (slopes) as well as the level after the lockdown was lifted (intercept) and integrated into the model for each well-being facet.

## Results

### Descriptive statistics

To illustrate the distribution of the mental well-being measures, the distribution across all measurement waves is shown in Fig 2. Since all distributions were skewed, the median is also included in the graph. The distributions of Stress and Psychological Strain at T1 and T5 exhibit a bimodal pattern and highlight the individual differences in the sample due to the broad range. Additionally, the captured well-being facets were correlated with one another. Because the facets are related to different components of well-being, only moderate correlations were found on average ($.11 \leq r \leq .37$, see S4 Table), with Stress and Psychological Strain being the exceptions with a strong positive correlation ($.70 \leq r \leq .82$, see S4 Table). Nevertheless, these two facets will also be analysed separately to maintain a consistent approach.

### Growth curve models for SWB facets

To test Hypothesis 1, four models for each well-being facet (Life Satisfaction, Stress, Psychological Strain, and Loneliness) were considered first. All model fit statistics can be found in Table 2. For all mental well-being facets, the best fitting model was identified. The results of the final best-fitting models including unstandardised means and variances of the intercepts and slopes, the correlations between them, as well as the loadings at each wave are presented in Fig 3 for Life Satisfaction and Loneliness and in Fig 4 for Stress and Psychological Strain.

**Life satisfaction.** The best model fit was found for the quadratic trend tested in Model 4. Individuals exhibited a positive trajectory towards the end of the lockdown, meaning that the

**Table 2. Model comparisons life satisfaction.**

|  | $\chi^2$ (df) | $p$ | CFI | RMSEA | SRMR | AIC | $\Delta\chi^2$ | df | $p$ |
|---|---|---|---|---|---|---|---|---|---|
| *Life Satisfaction* |  |  |  |  |  |  |  |  |  |
| Model 1 | 32.25 (14) | .004 | .964 | .067 | .070 | 10,652.71 | 10.21 | 4 | .037* |
| Model 2 | 39.7 (14) | < .001 | .950 | .080 | .085 | 10,660.16 | 7.45 | 0 |  |
| Model 3 | 21.88 (10) | .016 | .977 | .064 | .047 | 10,650.34 |  |  |  |
| **Model 4** | **16.44 (10)** | **.088** | **.987** | **.047** | **.029** | **10,644.89** | -5.44 | 0 |  |
| *Stress* |  |  |  |  |  |  |  |  |  |
| Model 1 | 25.93 (14) | .026 | .971 | .054 | .050 | 11,080.88 | 3.07 | 4 | .505 |
| Model 2 | 62.1 (14) | < .001 | .883 | .109 | .089 | 11,117.06 | 36.177 | 0 |  |
| **Model 3** | **12.13 (10)** | **.276** | **.995** | **.027** | **.033** | **11,075.09** |  |  |  |
| Model 4 | 22.86 (10) | .011 | .969 | .067 | .043 | 11,085.82 | 10.73 | 0 |  |
| *Psychological Strain* |  |  |  |  |  |  |  |  |  |
| Model 1 | 43.19 (14) | < .001 | .928 | .085 | .072 | 11,239.59 | 20.17 | 4 | < .001 |
| Model 2 | 48.62 (14) | < .001 | .915 | .092 | .080 | 11,245.02 | 5.43 | 0 |  |
| **Model 3** | **15.51 (10)** | **.114** | **.986** | **.044** | **.038** | **11,219.91** |  |  |  |
| Model 4 | 23.03 (10) | .011 | .968 | .067 | .042 | 11,227.42 | 7.51 | 0 |  |
| *Loneliness* |  |  |  |  |  |  |  |  |  |
| Model 1 | 3.91 (8) | .866 | 1.00 | .000 | .026 | 8,831.07 | 3.3 | 4 | .51 |
| Model 2 | 12.92 (8) | .115 | .98 | .048 | .058 | 8,840.09 | 9.02 | 0 |  |
| Model 3 | 2.42 (4) | .659 | 1.00 | .000 | .024 | 8,837.58 |  |  |  |
| **Model 4** | **0.61 (4)** | **.962** | **1.00** | **.000** | **.008** | **8,835.77** | -1.81 | 0 |  |

Notes. The $\chi^2$ column contains robust test statistics that should be reported per model. The basis for a robust difference test ($\Delta\chi2$) is a function of two standard (not robust) statistics, using the Satorra-Bentler-Method (2001).

lockdown had a negative effect on individuals' Life Satisfaction that vanished after the lockdown was lifted. As depicted in Fig 3, a positive standardised mean score ($\bar{x}_i = 71.4$; $p < .001$) was found. In comparison with the mean scores during the lockdown this indicates that individuals were indeed less satisfied during the lockdown and more satisfied with their life after the lockdown had ended. Additionally, significant variation between people's level of Life Satisfaction after the lockdown was identified ($s = 373.39$; $p < .001$). For the slope factors, both the linear trend s ($\bar{x}_s = 3.84$; $p < .001$) and the quadratic trend q ($\bar{x}_q = 0.75$; $p = .004$) were significant. The standard deviations of s and q were not significant, meaning that the people under study exhibited relatively similar trajectories. Meanwhile, the correlation of i and q was significant and negative. This indicates that people who did not suffer as much during the lockdown (Life Satisfaction did not drop as much) continued to exhibit higher values afterwards, while people who had suffered more continued to have lower values of Life Satisfaction after the lockdown.

**Stress.** For Stress, Model 3 provided the best fit. A significant positive standardised mean score for the intercept was found ($\bar{x}_i = 40.89$; $p < .001$). In comparison with the mean scores during the lockdown this suggested that people were more stressed after the end of the lockdown. However, there was also significant variation between respondents. With respect to the standardised slope parameters (s1 and s2) for stress, the model indicated a significant change in line with Slope 1 ($\bar{x}_{s1} = -8.74$, $p_{s1} < .001$). This means that the lockdown had a positive effect on reported Stress levels, which were then higher after the lockdown ended. Additionally, differences between participants concerning the trajectories of Stress were found for Slope 1 indicating that individuals reacted differently to the lockdown. For Slope 2, a negative correlation with the intercept was found, indicating that people with lower levels of Stress during the lockdown also had lower Stress levels after the lockdown had ended.

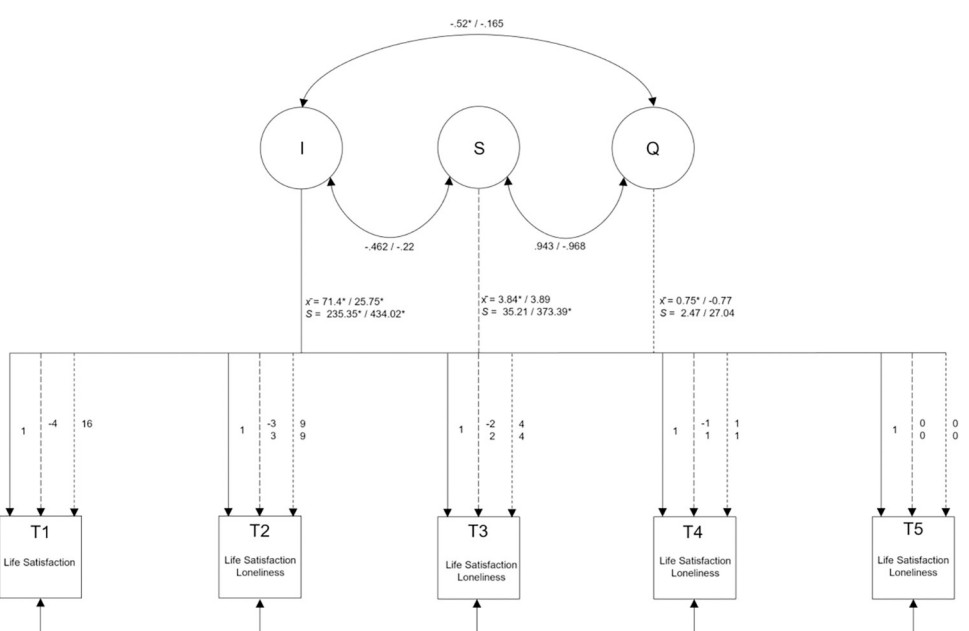

**Fig 3. Quadratic growth curve model for life satisfaction and loneliness.** The Interceptmeans ($\bar{x}$) can be interpreted as the average SWB at time 5 (after lockdown). $\bar{x}$ at S represents the linear slope factor and $\bar{x}$ q the quadratic one. The variances s indicates what individual variance is found in these components. The correlations between intercept and slope mean that the magnitude of the base level is related to the change.

**Psychological strain.** Psychological Strain showed a highly similar pattern of results to Stress, and the best fitting model was again Model 3. Again, a positive standardised mean score for the intercept was found ($\bar{x}_i$ = 44.97; $p$ < .001). That people were more psychologically strained after the lockdown can be assumed with regard on the mean scores. Moreover, the variation between participants' ending levels of Strain was significant. Concerning the trajectory, both slope parameters were found to be significant ($\bar{x}_{s1}$ = -7.91, $p_{s1}$ < .001; $\bar{x}_{s2}$ = 1.42, $p_{s2}$ = .014). Apparently, the lockdown led to a negative value for Slope 1, indicating a positive effect. This means the experienced strain level was lower during the lockdown than afterward. The positive value for Slope 2, in contrast, indicated a negative effect over time. Significant standard deviations for both slope parameters were found; therefore, it can be assumed that people differed in terms of their trajectories. Additionally, the intercept and Slope 2 were negatively correlated, meaning that people who experienced less change in Psychological Strain over time tended to have higher final values.

**Loneliness.** Like Life Satisfaction, the model with the best fit for Loneliness was Model 4. The standardised mean score for the intercept of Loneliness was significant ($\bar{x}_i$ = 25.75; $p$ < .001). In comparison with the mean scores during the lockdown, people felt less alone after the restrictions were lifted. Variation between people was also significant; however, no significant linear or quadratic slope was identified ($\bar{x}_s$ = 3.89, $p_s$ = .104; $\bar{x}_q$ = -0.77, $p_q$ = .285). Concerning the linear trend (s), a significant standard deviation, indicating variation between people regarding Loneliness, was shown.

## Conditional latent growth curve model including personality factors and Personality Organisation

To address Hypothesis 2, we expanded the previously defined LGCM by adding the Big Five personality factors, Personality Organisation as well as age (as a form of basic socio-

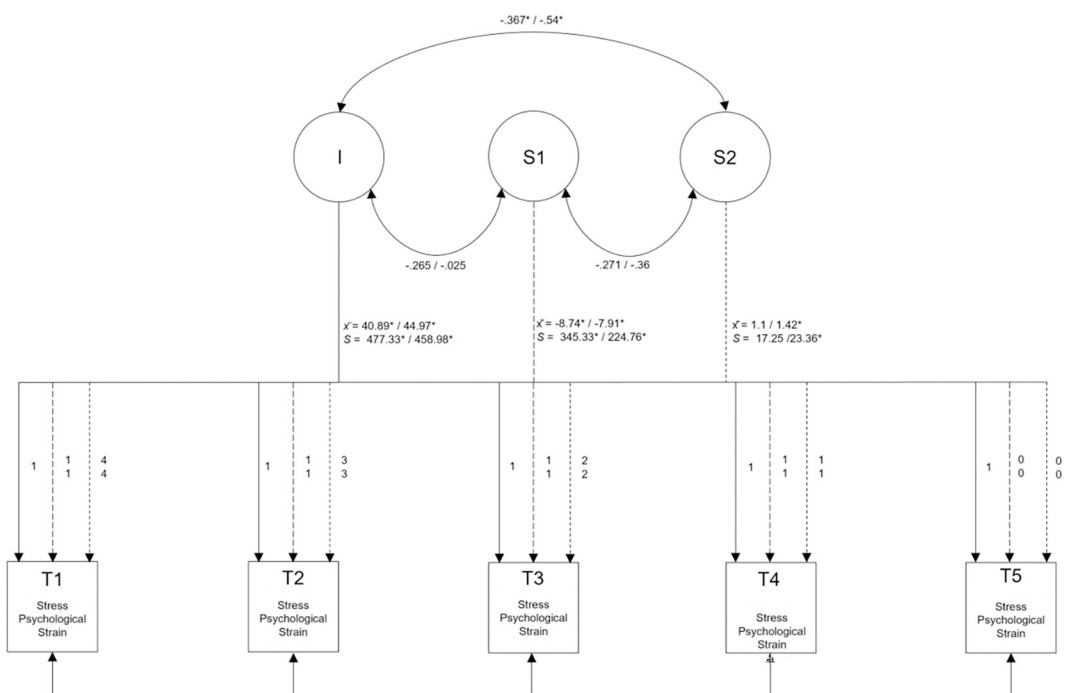

**Fig 4. Latent growth curve model for stress and psychological strain.** The Interceptmeans ($\bar{x}$) can be interpreted as the average SWB at time 5 (after lockdown). $\bar{x}$ at S1 represents the average rate of change from post-lockdown vs. lockdown and $\bar{x}$ S2 the average rate of change across lockdown. The variances s indicates what individual variance is found in these components. The correlations between intercept and slope mean that the magnitude of the base level is related to the change.

demographic information) as explanatory factors. Fit indices are provided in Table 3. For all models, the model fit was good. Estimates of the intercept and slope factors as well as path coefficients for the effects of personality on the trajectories and the level of mental well-being after the lockdown are shown in Table 4.

Focussing on the effect of personality, the results indicate personality factors were related to mental well-being to different degrees.

In particular, adding personality contributed to explaining individual differences in the trajectory of mental well-being in terms of Stress, Psychological Strain and Loneliness. Interindividual differences in mental well-being levels after the lockdown, however, could not fully be explained by personality, but some patterns were found.

Neuroticism was related to the average level of all mental well-being measures after the lockdown (see S1 Fig). Moreover, it had a negative association with the intercept of Life Satisfaction ($\bar{x}_i$ = -.7.21, $p$ < .001) and a positive association with the intercepts of Stress ($\bar{x}_i$ = 10.16, $p$ < .001), Psychological Strain ($\bar{x}_i$ = 11.6, $p$ < .05), and Loneliness ($\bar{x}_i$ = 5.27, $p$ < .05). As expected, people scoring higher on Neuroticism were on average less satisfied with their

**Table 3. Summary of model fit statistics for the conditional LGCMs including personality and age.**

| | $\chi^2$ (df) | $p$ | CFI | RMSEA | SRMR | AIC |
|---|---|---|---|---|---|---|
| Life Satisfaction | 33.65 (24) | .091 | .984 | .039 | .021 | 10,149 |
| Stress | 35.17 (24) | .066 | .977 | .042 | .025 | 10,590 |
| Psychological Strain | 29.7 (24) | .195 | .988 | .03 | .023 | 10,764 |
| Loneliness | 4.434 (11) | .955 | 1 | 0 | .01 | 8,435 |

**Table 4. Estimates for intercept and slope, effects of personality, sex and age based on conditional LGCM for all SWB measures.**

| Model | | $\bar{x}$ | s | E | A | C | N | O | IPO | age | $R^2$ |
|---|---|---|---|---|---|---|---|---|---|---|---|
| Life Satisfaction | i | 58.44* | 160.31* | -0.08 | 2.83 | 4.35 | -7.21** | 3.05 | -2.63 | 0.03 | .314 |
| | s | 3.59 | 28.75 | -2.39 | -0.858 | 3.52 | 0.48 | 0.71 | -0.93 | -0.11 | .258 |
| | q | 1.93 | 2.01 | -0.65 | -0.02 | 0.82 | 0.1 | -0.16 | -0.3 | -0.03 | .224 |
| Stress | i | 31.62 | 370.8* | 4.52* | -1.23 | -3.52 | 10.16** | -2.46 | -1.67 | -0.13 | .192 |
| | s1 | -11.69 | 233.81 | -8.8* | -1.92 | 10.68* | -1.15 | -1.68 | 4.48 | -0.1 | .268 |
| | s2 | -9.12 | 12.81 | 1.49* | -0.85 | -0.62 | 0.3 | 1.18 | 0.34 | 0.1* | .278 |
| Psychological Strain | i | -16.85 | 329.44* | 2.46 | 1.8 | -0.59 | 11.6* | 1.06 | 5.81 | 0.01 | .243 |
| | s1 | 14.43 | 151.51 | -5.77 | -0.12 | 8.56* | -0.62 | -4.32 | -3.03 | -0.27 | .301 |
| | s2 | -3.56 | 19.32* | 0.78 | -1.39 | -0.26 | -0.51 | 1.09 | 0.92 | 0.09* | .173 |
| Loneliness | i | 15.74 | 281.75* | 0.29 | -5.07* | -5.71* | 5.27* | 9.24* | 3.56 | -0.3* | .325 |
| | s | -10.94 | 332.96 | -2.88 | 6.24 | 5.36 | 3.26 | -7.93* | 0.47 | 0.14 | .137 |
| | q | 2.63 | 24.22 | 0.94 | -1.29 | -1.31 | -0.82 | 1.94 | -0.49 | -0.05 | .123 |

Notes.

*p < .05

**p < .001.

lives and at the same time more stressed, strained and lonelier when the lockdown ended compared to others. With respect to conscientiousness, significant differences were found concerning the level of Loneliness. Conscientious people felt less lonely after the lockdown than others ($\bar{x}_i$ = -5.71, $p < .05$). Additionally, our analysis revealed a positive effect for Conscientiousness on changes (s1) in Stress ($\bar{x}_{s1}$ = 10.68, $p < .05$) and Psychological Strain ($\bar{x}_{s1}$ = 8.56, $p < .05$), indicating differences in the trajectories of Stress and Strain in the sense that more conscientious participants reported more Stress and burden than others (S2 Fig). For the relation between Stress and Extraversion a positive association was found, indicating that people with higher Extraversion scores also had higher levels of Stress after the lockdown than during ($\bar{x}_i$ = 4.52, $p < .05$). Concerning the trajectories, contrasting effects were found, namely a negative association with Slope 1 ($\bar{x}_{s1}$ = -8.8, $p < .05$) and a positive association with Slope 2 ($\bar{x}^{Std}_{s2}$ = .29, $p < .05$), indicating that more extravert people tend to have less Stress during the lockdown (S3 Fig). In addition, Openness to experiences was negatively associated with the trajectory of Loneliness ($\bar{x}_s$ = -7.93, $p < .05$) as well positively related to reported Loneliness level after the lockdown ($\bar{x}_i$ = 9.24, $p < .05$). This indicated that individuals higher on Openness also reported higher levels of Loneliness after the lockdown had ended than others (S4 Fig). For Agreeableness, the opposite effect was found ($\bar{x}_i$ = -5.07, $p < .05$), as more agreeable people reported less Loneliness after the lockdown than others (S5 Fig). For Personality Organisation no associations were found with any mental well-being facet. However, age appeared to play a role. We found a positive association between age and the course of Stress ($\bar{x}^{Std}_{s2}$ = 0.1, $p < .05$) and Psychological Strain ($\bar{x}^{Std}_{s2}$ = 0.09, $p < .05$). In addition, age was negatively associated with reported Loneliness after the lockdown ($\bar{x}_i$ = -0.3, $p < .05$). Older people thus felt less alone.

In order to address group differences (Hypothesis 3), we added high-risk group status, essential workers as well as gender to the models utilized to test Hypothesis 1. The fit of these four models is reported in S5 Table and the estimated path coefficients in S6 Table. Overall, no differences between essential workers and the rest of the participants were found in terms of mental well-being. For high-risk group status, one difference in the experienced level of Loneliness emerged: People who believed they belong to this group differed from the rest in the trajectory of the slope factor ($\bar{x}_s$ = -18.17, $p < .05$). While Loneliness remained relatively stable

over the four survey points in the lower-risk group, the reported Loneliness level decreased steadily in the high-risk group. Different trajectories were found for women and men on Life Satisfaction ($\bar{x}_s$ = -5.07, $p < .05$), and gender was found to influence Stress levels after the lockdown had ended ($\bar{x}_i$ = -10.97, $p < .05$). On average, female participants stated that they were more stressed after the restrictions were lifted. The influence of age on the trajectory of Stress uncovered in the analyses addressing Hypothesis 2 was again evident ($\bar{x}_{s2}$ = 0.13, $p < .05$).

## Discussion

This study aimed to examine the impact of the first lockdown in Germany on different mental well-being facets (Life Satisfaction, Stress, Psychological Strain, and Loneliness). Because it was largely unknown in which direction and to what extent the lockdown had influenced well-being, the basic models analysed change trajectories in an exploratory way. Additionally, we studied the degree to which personality was associated with individual variability in people's mental well-being levels over time. Here, we focussed not just on the Big Five personality factors but also on a measure of mental health, namely personality organisation.

Applying a longitudinal approach over the course of the first lockdown, the study results support the set-point theory by showing that a decrease in Life Satisfaction was evident immediately after the onset of the lockdown, which disappeared after the lockdown was over. Surprisingly, personality traits did not lead to stronger or weaker reactions to the lockdown with respect to Life Satisfaction. However, if other components of mental well-being are considered, it becomes apparent that the influence of the lockdown is more complex. Contrary to the assumption that the lockdown led to a decline in mental well-being, we instead found significant reductions in both Stress and Psychological Strain, indicating an enhancement. Here, the Stress and Psychological Strain Levels increase again after the lockdown is lifted, i.e., showing more lasting effects. Additionally, as expected, people felt lonelier during the lockdown than afterwards.

### Life satisfaction

As expected, Life Satisfaction initially drops after the onset of the lockdown and then returns to the original value after the restrictions were lifted. This supports the set-point theory. As shown in previous studies, well-being often returns to the initial level after a short-term decrease or increase [9–11]. In particular, the increase found in our study immediately after the end of the lockdown suggests that it only had a short-term negative impact on Life Satisfaction. Thus, after the restrictions were relaxed, Life Satisfaction rose again, suggesting that individuals were happy to return to 'life the way it was'. This finding is in line with results from Foa and colleagues [52], who reported a drop in Life Satisfaction throughout the lockdown period with a return to (or even surpassing of) pre-lockdown levels afterwards.

Variation between participants, on the other hand, were only found with regard to the final level of Life Satisfaction, but not the course of Life Satisfaction. Apparently, the initial drop in Life Satisfaction was comparable for everybody, which is less surprising given that the COVID-19 pandemic represented an unprecedented situation for all. The impact was highest right at the beginning and started to fade rapidly over the course of the following weeks. Therefore, we can assume that the level after lockdown is the usual Life Satisfaction level. People who tended to be more satisfied before the lockdown also showed a flatter curve, i.e., they were better able to cope with the situation or even made good use of the lockdown.

As expected and well-known in the literature, Neuroticism showed an effect on the intercept of Life Satisfaction, i.e., higher neuroticism was associated with lower Life Satisfaction after restrictions were lifted. In line with a set-point perspective, this may in part reflect a

return to Life Satisfaction differences associated with Neuroticism before the pandemic. In their meta-analyses, Anglim and colleagues [53] reported a moderate correlation between Neuroticism and Life Satisfaction ($r$ = -.39; [-.41 $\leq$ CI$_{95\%}$ $\leq$ -.38]), which was larger than our post-lockdown effect size ($r$ = -.19). In the absence of pre-pandemic data in this study, we cannot say whether the effect found in our sample is equal to or greater than the difference found in the pre-lockdown period. However, it can be concluded that the difference between neurotic and less neurotic individuals remains. Previous research on Life Satisfaction in the context of the COVID-19-pandemic has shown that neurotic people paid more attention to information concerning the pandemic and worried about it more [36, 37] or had a more negative appraisal of the pandemic [35]. Nevertheless, these differences tied to Neuroticism may not necessarily influence post-lockdown Life Satisfaction, as the differences were also present before the pandemic.

There were gender differences concerning the trajectories of Life Satisfaction. We found that women exhibited a flatter trajectory during the lockdown and tended to have lower levels of Life Satisfaction afterwards. This finding is in line with the fact that mothers' well-being was particularly negatively affected by the pandemic and with more mental health problems for women in general [18, 39].

## Stress

Our study reported an increase in Stress levels when the restrictions were relaxed. This is in line with reports that Stress levels decreased substantially in the first month of the lockdown [54]. It is possible that for many, the lockdown was seen as an opportunity to "slow down". Many social obligations and appointments were suspended. A look at the mean values across survey waves reveals that the Stress reduction is particularly evident from T2 onwards, possibly reflecting that by then, a possible sense of overwhelming uncertainty and far-reaching change had given way to a more adaptive reaction. However, participants differed in their final level of Stress and whether their Stress was reduced during the lockdown. A flatter Stress trajectory during the lockdown was associated with higher Stress levels after the lockdown, suggesting that participants with no reduction in Stress during the lockdown or for whom the lockdown had only a weak influence had higher Stress values after the lockdown compared to participants who experienced large reductions in Stress. Adding personality and age to the regression, more variance could be explained, supporting the important role of personality differences as sources of individual differences in mental well-being.

In our study, the three personality dimensions most relevant for Stress were Extraversion, Conscientiousness, and Neuroticism which mirrors both pre-pandemic meta-analytic results [53] as well as findings during the COVID-19 pandemic [55]. In our study, participants scoring higher on Extraversion had higher post-lockdown Stress levels as well as a greater reduction in Stress during the lockdown. This suggests that those individuals experienced greater differences between the lockdown and post-lockdown periods in terms of their Stress levels. It seems plausible that the decreased social obligations had a larger effect on extraverts, who tend to have more social contacts and potentially more scheduling stress compared to introverts. For this reason, it is conceivable that introverts' Stress levels remained quite stable, whereas for extraverts, a return to a busier and more stressful schedule took place after the lockdown was lifted.

Another personality dimension related to Stress level was Conscientiousness. In general, conscientious people are well organised, reliable, and prudent. During the first lockdown, a number of additional restrictions and recommendations were made (e.g., contact reduction, increased cleaning and disinfection). The increased Stress levels during the first lockdown

could therefore have resulted from particularly conscientious individuals taking the restrictions and recommendations seriously; thus, for example, the shortage of disinfectants may have led to increased Stress among this group. In particular, this could be due to the fact that conscientious people are usually less spontaneous and thus slower to adapt to new situations [56]. Nevertheless, particularly conscientious individuals perceived a higher level of efficacy to prevent COVID-19 [55]. The same study found no correlation between Conscientiousness and Stress for their participants in May 2020 after perceived efficacy to prevent COVID-19 was included as a mediator. Potentially, conscientious individuals engaged in more protective procedures during the first months.

In contrast Neuroticism, was associated with the post-lockdown level in May 2020. Neurotic people were more stressed after the lockdown. This positive association between Neuroticism and Stress was also found by Liu and colleagues [55]. Generally speaking, this result falls in line with Neuroticism's association with negative appraisals and the tendency to ruminate and worry more about the COVID-19 pandemic among individuals scoring high on Neuroticism [36, 37].

Furthermore, significant differences between men and women emerged. On average, women had higher Stress levels during the lockdown. Additionally, different trajectories depending on age were found. Older participants appeared to experience less Stress reduction than younger ones. Having young children at home constituted an additional burden, as parents often not only worked from home themselves, but also had to look after and/or home-school their children [57]. Moreover, studies suggest that childcare often remains the task of women [39, 58]. Therefore, higher values on Stress for women and older participants can potentially be explained by different responsibilities (e.g., childcare, household, job, income).

## Psychological strain

The results concerning Psychological Strain are comparable with the ones for Stress. Here too, participants had a lower level of Psychological Strain during the lockdown; therefore, it can be argued that the lockdown had some positive effect. The linear trajectory shows that the reduction in Strain followed a week-by-week trend. Like Stress, Psychological Strain can be regarded as a form of negative affect; thus, our findings align with those reported by Modersitzki and colleagues [54].

Additionally, it was shown that a flatter trajectory led to a higher Strain level after the lockdown. People who calmed down less during the lockdown were more strained afterwards. This shows that it may have been important to organise and utilise the lockdown as positively as possible or to use various coping strategies (e.g., task-oriented coping, emotion-based coping, or distraction), for reducing Psychological Strain as well as Stress [59].

Similarly to Stress, Conscientiousness and Neuroticism were significant regressors, whereas Extraversion was not associated with Psychological Strain. Neuroticism was also associated with the level of Strain after the lockdown. Again, people scoring higher on this dimension were more strained during the last survey wave, in line with the fact that Neuroticism and negative appraisal are associated with one another [36, 37]. Additionally, more conscientious individuals experienced more Psychological Strain during the lockdown. This is in line with Venkatesh and colleagues [60] who reported that more conscientious workers also reported higher Strain and lower Satisfaction levels during the pandemic.

In contrast to the mental well-being facets discussed earlier, no group differences in Psychological Strain were found. This might be because Psychological Strain is influenced by other factors that became especially relevant during the COVID-19 pandemic. In addition to the previously mentioned coping behaviours, certain framework conditions during the

lockdown could have been decisive. For example, it is conceivable that people without children were less stressed, as were those who did not have to worry about losing their jobs. Finally, socio-economic status could play a role as well, because it has been generally associated with better well-being during the pandemic [18, 40].

## Loneliness

Generally, people had lower Loneliness scores after the lockdown compared to during the lockdown. However, even though Loneliness levels after the lockdown seemed to be lower than during it, they could still be higher than in the previous years. This is supported by data from the German SOEP, which found an increase Loneliness levels in April 2020 compared to those reported in previous years [61]. In contrast to the SOEP data, we inspected Loneliness levels over a shorter period of time, but with higher frequency (i.e., weekly surveys) more closely. The participants in our study also exhibited increased Loneliness levels in April 2020, although these decreased significantly after the restrictions were lifted. Similar findings exist for a US sample [62]. A possible reason for the steadily decreasing Loneliness scores in our study during the lockdown could be due to alternative forms of digital communication or increasing engagement in digital activities to keep in touch with family and friends [63]. The lower Loneliness scores after the lockdown can be explained by the fact that people met up more after the restrictions had been lifted, albeit potentially still less than normally.

Moreover, people varied in their linear time trends as well as their post-lockdown levels of Loneliness. Adding the Big Five as well as Personality Organisation to the regression analyses showed that more agreeable and conscientious people had lower levels of Loneliness, more neurotic and open individuals had higher levels. The differences found for Agreeableness, Conscientiousness and Neuroticism are consistent with those from the meta-analysis by Buecker and colleagues [64]. These authors also showed that differences in Loneliness prior to the COVID-19 pandemic were likewise related to Agreeableness ($r$ = -.27, [-.3 $\leq$ CI$_{95\%}$ $\leq$-.23]), Conscientiousness ($r$ = -.22, [-.25 $\leq$ CI$_{95\%}$ $\leq$-.19]), and Neuroticism ($r$ = .39, [.35 $\leq$ CI$_{95\%}$ $\leq$ .43]) in the respective directions. The effects found in our study for people with higher levels of Neuroticism and Conscientiousness were smaller than those reported in the former study. However, our data collection took place during the first lockdown, so it is possible that the influence of personality was weakened during this period because the pandemic seemed so overwhelming. Due to this extraordinary situation, other situational or individual factors (e.g., working from home, childcare/home-schooling, living conditions, socio-economic-status) might have also been relevant. Nevertheless, a significant effect was also found for the Openness dimension. While the pre-pandemic meta-analysis reported a small negative association ($r$ = -.12, [-.15 $\leq$ CI$_{95\%}$ $\leq$ -.09]), we found a positive correlation between Openness and Loneliness ($r$ = .12), possibly because people who are motivated to seek out new experiences were more affected by the lack of social interaction and the contact restrictions during the lockdown. Our finding that Extraversion was not associated with Loneliness during the pandemic is in line with the literature [34]. On the other hand, individuals with higher Agreeableness and, in particular, Conscientiousness scores may be more inclined to abide by and agree with pandemic-related rules, and their assessment that the measures are necessary may help to prevent them from developing intense feelings of Loneliness.

Age and risk group status seemed to be relevant as well. Younger people and those who do not identify themselves as being in the high-risk group felt lonelier than the elderly during the lockdown. However, it can be stated that the lockdown, which was introduced to protect the old, vulnerable generations, has placed a heavy burden on the younger generations in particular. This finding is consistent with other studies during the pandemic [61, 65].

Taken together, our study clearly reveals that individual differences, including Big Five personality traits as well as age and gender, explained how individuals react to the pandemic and its restrictions in terms of their mental well-being.

## Limitations

This raises three questions. To answer the first one, future studies should focus on other influencing factors. It is possible that coping behaviour during the lockdown can be protective to a certain degree [59]. Therefore, differences as well as changes in coping strategies during the lockdown should be analysed. Additionally, one's evaluation of the situation as a whole or the restrictions can also be relevant for mental well-being. It is conceivable, for example, that people who considered the restrictions necessary were able to cope better with the situation and generally assess it more positively compared to individuals who disagreed with the measures taken. Additionally, it should be considered that personality might change in response to the COVID-19 pandemic. Along these lines, Sutin and colleagues [66] report a decrease in Neuroticism scores during the first six weeks of the pandemic. Their interpretation of this change was that people attribute their feelings of anxiety and distress more on the pandemic and less on their own personality. In our study, we also found a significant decrease in Neuroticism between the first and the last survey. Nevertheless, the LGCM results remained unchanged when Neuroticism scores from the last survey were included.

The third limitation of this study is the operationalisation of SWB. In our study, we mainly focussed on mental well-being and Life Satisfaction, as the main cognitive factor of SWB. Certainly, SWB is a wide-ranging construct, and more detailed measurements of different facets could be interesting. Fear, tedium, as facets of negative affect or also facets of positive affect, like optimism or happiness, might be associated with individual differences in dealing with the COVID-19 pandemic. To additionally test set-point theory, SWB before and after a lockdown period should also be assessed. In our study, we were not able to measure levels of well-being at those time points. For Loneliness, in particular, data collection began only during the second survey wave. Therefore, the first week of the lockdown was not included when modelling the trajectory.

## Future perspective

It could also be interesting for future research to focus on Personality Organisation. It turned out that scores improved after the lockdown, contrary to our assumption that the lockdown would have a negative impact on mental health. The mean difference showed that the risk of mental illness decreased across all participants. This is related to our finding that the trajectory of mental well-being reversed, and people became less stressed. This, in combination with the post-lockdown rise of stress and psychological strain, suggests that the first lockdown was a kind of break for many people and that was therefore at least partly beneficial to them. As previously mentioned, our study took place following the beginning of the pandemic, so the reported post-lockdown values were collected during the transition to a further pandemic phase. Nevertheless, Personality Organisation seemed to be independent of the measured well-being facets and changes in it over time. Thus, it might be well-suited for measuring the psychological impact of the pandemic. It would also be interesting to see whether the reduction of structural impairment is moderated by personality or other individual factors.

The influence of the pandemic or individual mitigation measures could be more comprehensively analysed using panel data. In this way, mental well-being and the influence of further lockdowns on it could be examined throughout the entire course of the pandemic. It would be particularly interesting to see whether set-point theory applies only to single lockdown periods

or to the pandemic as a whole. Previous research on set-point theory, but also our study on the influence of the first lockdown, allow both assumptions. In addition, the relation between different mental well-being measures (e.g., Stress, Loneliness) and SWB facets (e.g., Life Satisfaction) as well as the moderation of this relationship through personality should also be investigated. Getzmann and colleagues [67] found a moderating effect of Agreeableness on the relation between Feeling of Missing and changes in Stress in terms of higher Agreeableness leading to a stronger association of Feeling of Missing and Stress, measured both pre-COVID-19 and after the lockdown. It is therefore possible, that other personality facets, which did not seem relevant in our study, exert their effect through moderation processes with well-being and related stress measures.

Especially because of the global nature of the COVID-19 pandemic, it is important to replicate results in other samples and in other countries. In addition, since the pandemic has been ongoing for more than two years now, the impact of the first lockdown (or further lockdowns) should also be analysed in longer follow-up designs. In this way, long-term effects of the lockdowns can be investigated and action recommendations with regard to potential future pandemic developments can be derived. It is conceivable that influences of personality facets and the group differences we found in our study will continue to be important for mental health, coping and dealing with the pandemic. Again, further analyses of various panel data may be helpful in this regard. This view is supported by the fact that some longitudinal surveys (e.g., SOEP [68], SHARE [69]) have included additional interviews that specifically address the COVID-19 pandemic and related well-being.

## Conclusion

The study explored the effects of the first lockdown on mental well-being in Germany. The results suggest that Life Satisfaction exhibited short-term changes as a reaction to the lockdown in response to the COVID-19 pandemic. Contrary to our hypotheses, the lockdown actually reduced Stress and Psychological Strain after the second week. In addition, even Loneliness scores improved during the course of the lockdown. When adding personality characteristics and other potential influencing factors, our results showed that Neuroticism and Conscientiousness were the two dimensions associated most with mental well-being during the first month of the pandemic. Neuroticism was negatively associated with the final level of all well-being facets under study, while Conscientiousness was positively associated with the trajectories of Stress and Psychological Strain. Additionally, age and gender also seemed relevant, but mainly for the course of mental well-being. Our research suggests that restrictions in response to the pandemic can have a short-term influence on all mental well-being measures and that personality traits should be considered when analysing psychological well-being.

## Supporting information

**S1 Table. Comparison complete cs. incomplete responders.**
(PDF)

**S2 Table. Correlations between all measured personality facets.**
(PDF)

**S3 Table. Means and standard deviations of SWB.**
(PDF)

**S4 Table. Correlations of SWB facets over time.**
(PDF)

**S5 Table. Summary of model fit statistics for the conditional LGCM including risk group, systemic relevance, sex and age.**
(PDF)

**S6 Table. Estimations of intercept and slope, effects of group differences, sex and age.**
(PDF)

**S1 Fig. The trajectory of all SWB facets for more and less neurotic participants.**
(TIF)

**S2 Fig. The trajectory of Stress, Psychological Strain and Loneliness for more and less conscientious participants.**
(TIF)

**S3 Fig. The trajectory of Stress for extrovert and introvert participants.**
(TIF)

**S4 Fig. The trajectory of Loneliness for open and less open participants.**
(TIF)

**S5 Fig. The trajectory of Loneliness for high and low agreeable participants.**
(TIF)

## Author Contributions

**Conceptualization:** Julie Levacher, Elisabeth Hahn.

**Data curation:** Julie Levacher, Elisabeth Hahn.

**Formal analysis:** Julie Levacher, Elisabeth Hahn.

**Project administration:** Julie Levacher, Elisabeth Hahn.

**Supervision:** Frank M. Spinath, Elisabeth Hahn.

**Visualization:** Julie Levacher.

**Writing – original draft:** Julie Levacher.

**Writing – review & editing:** Julie Levacher, Frank M. Spinath, Nicolas Becker, Elisabeth Hahn.

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
