## [Decision Letter · Decision Letter 0]

20 Sep 2022

PONE-D-22-21116How did the beginnings of the global COVID-19 pandemic affect mental well-being?PLOS ONE

Dear Dr. Levacher,

Thank you for submitting your manuscript to PLOS ONE. After careful consideration, we feel that it has merit but does not fully meet PLOS ONE’s publication criteria as it currently stands. Therefore, we invite you to submit a revised version of the manuscript that addresses the points raised during the review process.

We look forward to receiving your revised manuscript.

Kind regards,

Alejandro Vega-Muñoz, Ph.D.

Academic Editor

PLOS ONE

Journal Requirements:

2. For studies reporting research involving human participants, PLOS ONE requires authors to confirm that this specific study was reviewed and approved by an institutional review board (ethics committee) before the study began. Please provide the specific name of the ethics committee/IRB that approved your study, or explain why you did not seek approval in this case.

Additional Editor Comments:

Dear Author, based on the comments of all reviewers my decision is that your work requires a major revision.

Please consider the comments of all reviewers, to elaborate your response and submit an improved version of your manuscript.

Reviewers' comments:

Reviewer's Responses to Questions

**Comments to the Author**

1. Is the manuscript technically sound, and do the data support the conclusions?

Reviewer #1: Yes

Reviewer #2: Yes

Reviewer #3: Yes

2. Has the statistical analysis been performed appropriately and rigorously? 

Reviewer #1: Yes

Reviewer #2: Yes

Reviewer #3: Yes

3. Have the authors made all data underlying the findings in their manuscript fully available?

Reviewer #1: No

Reviewer #2: No

Reviewer #3: Yes

4. Is the manuscript presented in an intelligible fashion and written in standard English?

Reviewer #1: Yes

Reviewer #2: Yes

Reviewer #3: Yes

5. Review Comments to the Author

Reviewer #1: I like this study, but the discussion suffers from a neglect of a highly relevant recent study (Getzmann et al., COVID-19 Pandemic and Personality: Agreeable People Are More Stressed by the Feeling of Missing, Int. J. Environ. Res. Public Health 2021, 18(20), 10759). This longitudinal study from Germany compared - in the same sample- stress levels that were obtained pre-Corona to post-Corona stress levels and related these changes to personality (Big 5) that was also measured pre-Corona. The results partly confirm the findings of the present study but also reported patterns that clearly diverge from the present findings (esp. wth respect to the relation of agreeableness and loneliness). I find it mandatory to consider this study in the discussion of the present paper.

Reviewer #2: This study is an investigation of mental health changes in response to the COVID-19 crisis. The sample, though relatively small, is followed up to 5 times over the first months of the pandemic in Germany. The manuscript is well-written and the authors clearly specify their research questions. However, there are a few aspects of the study that undermined my enthusiasm.

Majors:

1. Your terminology and definition of concepts. The authors refer to SWB (subjective well-being), but what they measure are different components of mental or psychological health. Per theory, SWB has three components: life satisfaction (LS), positive affect (PA), and negative affect (NA) (Andrews & Withey, 1976). Individuals are said to have high SWB if they experience LS and frequent PA (e.g., joy, optimism) and infrequent NA (e.g., sadness, anger). The authors only measure LS, and include measures of psychological strain, stress and loneliness, which in my view are constructs that are correlated but differ from subjective well-being. The authors noted this a limitation of their study (p. 37), but I would rather recommend them to revise their language and terminology in the manuscript and refer to mental health or mental well-being (as similarly done in the Measure section, p. 16).

2. Personally, I share the urgency to understand the impact of the COVID-19 pandemic on mental health, across different at-risk groups (for example, essential workers). However, in the introduction, a better job could be done to clearly state what is the unique contribution of this work. Moreover, one missing piece in the published literature is to understand long-term consequences of the pandemic. Have the authors considered to pair their data with other public available COVID-19 surveys? For example, SHARE (http://www.share-project.org/home0.html) includes similar mental health measures from a subsample from Germany, which might make possible to compare COVID-19 consequences on mental well-being, short-term and over 1 year, in the same country.

3. In term of analytical approach, the models seem adequately specified. However, for the model including personality as predictor of mental health changes, I would recommend to each personality measure in separate model and have the model with all traits entered together as a follow-up analysis.

4. In the discussion, please consider that personality might change as well in response to the COVID-19, as supported by recent evidence published in Plos One (Sutin et al. 2020, DOI: 10.1371/journal.pone.0237056; Sutin et al. in press).

5. Further, under study limitation (or direction for future studies), more stress should be placed on the need to replicate study results in future work and the use of longer follow-up (related to point 2).

Minors:

6. Abstract, the authors do not include any information on sample size, study design nor sample characteristics (for example, age).

Reviewer #3: I think it is a very interesting research, of great relevance to the context of the Covid-19 pandemic, and it is very well written. However, I recommend in the abstract to briefly indicate the methodology, and in the article in general, to improve the concordance between the concepts presented and what was measured, because although it talks about subjective well-being, which in the same introduction, it is mentioned that it is made up of the components life satisfaction, positive affect and negative affect, not all of these are measured.

6. PLOS authors have the option to publish the peer review history of their article (what does this mean?). If published, this will include your full peer review and any attached files.

Reviewer #1: No

Reviewer #2: No

Reviewer #3: No

---

## [Author Response · Author response to Decision Letter 0]

2 Nov 2022

Dear Dr. Vega-Muñoz,

Thank you very much for your comments on our manuscript titled “How did the beginnings of the global COVID-19 pandemic affect mental well-being?” (PONE-D-22-21116). We are also grateful for the reviewers’ comments and have now revised the paper in accordance with this advice. The revi-sions we made are highlighted in the file labelled 'Revised Manuscript with Track Changes'.

Below you can find the requested point-by-point responses in which we address all the concerns raised by you and the reviewers. For reasons of simplicity, we answered comments addressing the same issues together. We hope that we were able to reconcile all the concerns in an acceptable way, and we are looking forward to hearing from you again.

Yours sincerely,

XXX (names omitted due to blind review)

Reviewer #1: I like this study, but the discussion suffers from a neglect of a highly relevant recent study (Getzmann et al., COVID-19 Pandemic and Personality: Agreeable People Are More Stressed by the Feeling of Missing, Int. J. Environ. Res. Public Health 2021, 18(20), 10759). This longitudinal study from Germany compared - in the same sample- stress levels that were obtained pre-Corona to post-Corona stress levels and related these changes to personality (Big 5) that was also meas-ured pre-Corona. The results partly confirm the findings of the present study but also reported patterns that clearly diverge from the present findings (esp. wth respect to the relation of agreea-bleness and loneliness). I find it mandatory to consider this study in the discussion of the present paper.

R#1 A: Thank you for drawing our attention to this study. We found it interesting and helpful, espe-cially because of the analysed relationship between changes in stress and the feeling of missing out. The moderation found in this study, should – in our option –also be considered. Therefore, we included the study in our discussion (future perspective paragraph) instead of the introduction for two reasons. Including this study in our introduction creates the impression that this study could have influenced our pre-registered hypotheses, thus portraying our research process in a poten-tially misleading way. Second, the measures used by Getzmann and colleagues differed substantial-ly from our measures (e.g., their pre-covid measure spanned 22.5 months, resulting in very differ-ent pre-post-test intervals) which limits the transferability to our findings. Nonetheless, we agree that it is important to report the findings of this study which can inspire future research projects. Therefore, we added the following section to our manuscript:

“In addition, the relation between different mental well-being measures (e.g., Stress, Loneliness) and SWB facets (e.g., Life Satisfaction) as well as the moderation of this relationship through per-sonality should also be investigated. Getzmann and colleagues (66) found a moderating effect of Agreeableness on the relation between Feeling of Missing and changes in stress in terms of higher agreeableness leading to a stronger association of feeling of missing and stress, measured both pre-COVID-19 and after the lockdown. It is therefore possible, that other personality facets, which did not seem relevant in our study, exert their effect through moderation processes with well-being and related stress measures.”

Reviewer #2: This study is an investigation of mental health changes in response to the COVID-19 crisis. The sample, though relatively small, is followed up to 5 times over the first months of the pandemic in Germany. The manuscript is well-written and the authors clearly specify their research questions. However, there are a few aspects of the study that undermined my enthusiasm.

Majors:

1. Your terminology and definition of concepts. The authors refer to SWB (subjective well-being), but what they measure are different components of mental or psychological health. Per theory, SWB has three components: life satisfaction (LS), positive affect (PA), and negative affect (NA) (Andrews & Withey, 1976). Individuals are said to have high SWB if they experience LS and frequent PA (e.g., joy, optimism) and infrequent NA (e.g., sadness, anger). The authors only measure LS, and include measures of psychological strain, stress and loneliness, which in my view are constructs that are correlated but differ from subjective well-being. The authors noted this a limitation of their study (p. 37), but I would rather recommend them to revise their language and terminology in the manuscript and refer to mental health or mental well-being (as similarly done in the Measure sec-tion, p. 16).

R#2 A1: We fully agree that we did not capture all aspects of SWB in our measurement. Although SWB is measured in many studies by the main indicator of life satisfaction and we thus chose this designation, we understand the objection. Therefore, we revised our language and terminology and now refer to mental well-being instead. However, the limitations about SWB addressed in the manuscript remain, because we find it worth noting that other aspects such as fear, anxiety, and happiness should also be taken into account.

2. Personally, I share the urgency to understand the impact of the COVID-19 pandemic on mental health, across different at-risk groups (for example, essential workers). However, in the introduc-tion, a better job could be done to clearly state what is the unique contribution of this work. More-over, one missing piece in the published literature is to understand long-term consequences of the pandemic. Have the authors considered to pair their data with other public available COVID-19 sur-veys? For example, SHARE (http://www.share-project.org/home0.html) includes similar mental health measures from a subsample from Germany, which might make possible to compare COVID-19 consequences on mental well-being, short-term and over 1 year, in the same country.

R#2 A2: In order to illustrate the unique contribution of our work to COVID-19 related research more clearly, we have added the following section:

“In contrast to other studies, we placed a strong emphasis on the analysis of the trajectory of men-tal well-being by administering questionnaires at short intervals of weekly surveys during the lock-down. This approach allows for a closer inspection of changes throughout this phase of the pan-demic.”

We fully share your opinion that the combination of datasets and multiple-lab approaches benefit psychological research. Upon closer inspection of the SHARE data as well as further datasets (such as SOEP), we noticed too many substantial differences that render direct comparisons difficult without straying too far from the original research idea that was pre-registered. To further empha-size the importance of replication and combination with other data sets, however, we have includ-ed the following paragraph:

“Especially because of the global nature of the COVID-19 pandemic, it is important to replicate re-sults in other samples and in other countries. In addition, since the pandemic has been ongoing for more than two years now, the impact of the first lockdown (or further lockdowns) should also be analysed in longer follow-up designs. In this way, long-term effects of the lockdowns can be inves-tigated and action recommendations with regard to potential future pandemic developments can be derived. It is conceivable that influences of personality facets and the group differences we found in our study will continue to be important for mental health, coping and dealing with the pandemic. Again, further analyses of various panel data may be helpful in this regard. This view is supported by the fact that some longitudinal surveys (e.g., SOEP, SHARE) have included additional interviews that specifically address the COVID-19 pandemic and related well-being.”

3. In term of analytical approach, the models seem adequately specified. However, for the model including personality as predictor of mental health changes, I would recommend to each personali-ty measure in separate model and have the model with all traits entered together as a follow-up analysis. 

R#2 A3: Thank you for this suggestion. To test this approach, we set up each model including only one personality measure. Those resulted in a very similar pattern, even though some numerical values were a little bit higher (which could have been expected given the fact that the shared vari-ance between personality facets is not represented in this approach). Nevertheless, including per-sonality measures separately did not lead to substantial changes in the results and have therefore been omitted from the manuscript. Furthermore, since the separate models do not account for the variance shared between each personality measure, the separately including procedure may yield somewhat biased estimates. Therefore, we decided to hold on to the reported models including all five personality measures as well as their intercorrelations simultaneously. To justify this decision to the readers we added the following to our analytic strategy section:

“To analyse the combined effect of personality, all personality factors (Big Five and Personality Or-ganisation) were modelled simultaneously, while controlling for age. This approach allows us to estimate the effect of each personality factor while controlling for the effect of all other factors.”

4. In the discussion, please consider that personality might change as well in response to the COVID-19, as supported by recent evidence published in Plos One (Sutin et al. 2020, DOI: 10.1371/journal.pone.0237056; Sutin et al. in press).

R#2 A4: Thank you for raising this important issue. We added the study from Sutin and colleagues (2020) to our discussion. Additionally, we analysed differences in our measured personality facets from the first survey and the fifth. In our sample, significant differences were found for Neuroti-cism, indicating lower values after the lockdown (M1st = 2.88, SD1st = 0.8; M5th = 2.74, SD5th = 0.84, t(209) = 3.31, p = .001, d = .234). Even if the effect we found in our sample is very small, our data support the interpretation by Sutin and colleagues in the sense that people seem to attribute their worries to some degree on the COVID-19 pandemic rather than on their own personality now. To investigate if those changes lead to different results in our LGCM, we further explored the model estimations including the Neuroticism-score from the last survey period. Since this adjustment did not change our results, we added the following:

“Additionally, it should be considered that personality might change in response to the COVID-19 pandemic. Along these lines, Sutin and colleagues (65) report a decrease in Neuroticism scores dur-ing the first six weeks of the pandemic. Their interpretation of this change was that people attribute their feelings of anxiety and distress more on the pandemic and less on their own personality. In our study, we also found a significant decrease in Neuroticism between the first and the last survey. Nevertheless, the LGCM results remained unchanged when Neuroticism scores from the last survey were included.”

5. Further, under study limitation (or direction for future studies), more stress should be placed on the need to replicate study results in future work and the use of longer follow-up (related to point 2).

R#2 A5: see R#2 A2

Minors:

6. Abstract, the authors do not include any information on sample size, study design nor sample characteristics (for example, age).

R#2 A6: We added information concerning our analysed sample as well as a short summary of the methodology used.

“[…] Overall, 272 adults (Mage= 36.94, SDage= 16.46; 68.62 % female, 23.45 % male, 0.69 % non-binary) took part in our study with four weekly surveys during the lockdown as well as a follow-up one month after restrictions were lifted. To analyse the development of mental well-being during and shortly after the first lockdown in Germany latent growth curve models (LGCM) were calculat-ed. The considered facets of well-being differ by their trajectory. […]”

Reviewer #3: I think it is a very interesting research, of great relevance to the context of the Covid-19 pandemic, and it is very well written. However, I recommend in the abstract to briefly indicate the methodology, and in the article in general, to improve the concordance between the concepts presented and what was measured, because although it talks about subjective well-being, which in the same introduction, it is mentioned that it is made up of the components life satisfaction, posi-tive affect and negative affect, not all of these are measured.

R#3 A: Thank you for your comment on the abstract. We now briefly indicate the methodology in our abstract (see Reviewer #2, comment 6). We agree that we did not have a well-suited name for our measured well-being facets. Therefore, we followed the suggestion of Reviewer #2 to revise our language and terminology in the manuscript. We now define our measured components as mental well-being.

---

## [Decision Letter · Decision Letter 1]

14 Dec 2022

How did the beginnings of the global COVID-19 pandemic affect mental well-being?

PONE-D-22-21116R1

Dear Dr. Levacher,

We’re pleased to inform you that your manuscript has been judged scientifically suitable for publication and will be formally accepted for publication once it meets all outstanding technical requirements.

Kind regards,

Alejandro Vega-Muñoz, Ph.D.

Academic Editor

PLOS ONE

Reviewers' comments:

Reviewer's Responses to Questions

**Comments to the Author**

1. If the authors have adequately addressed your comments raised in a previous round of review and you feel that this manuscript is now acceptable for publication, you may indicate that here to bypass the “Comments to the Author” section, enter your conflict of interest statement in the “Confidential to Editor” section, and submit your "Accept" recommendation.

Reviewer #1: All comments have been addressed

Reviewer #3: All comments have been addressed

2. Is the manuscript technically sound, and do the data support the conclusions?

Reviewer #1: (No Response)

Reviewer #3: Yes

3. Has the statistical analysis been performed appropriately and rigorously? 

Reviewer #1: (No Response)

Reviewer #3: Yes

4. Have the authors made all data underlying the findings in their manuscript fully available?

Reviewer #1: (No Response)

Reviewer #3: Yes

5. Is the manuscript presented in an intelligible fashion and written in standard English?

Reviewer #1: (No Response)

Reviewer #3: Yes

6. Review Comments to the Author

Reviewer #1: (No Response)

Reviewer #3: (No Response)

7. PLOS authors have the option to publish the peer review history of their article (what does this mean?). If published, this will include your full peer review and any attached files.

Reviewer #1: No

Reviewer #3: No

---

## [Editor Report · Acceptance letter]

6 Jan 2023

PONE-D-22-21116R1 

How did the beginnings of the global COVID-19 pandemic affect mental well-being? 

Dear Dr. Levacher:

I'm pleased to inform you that your manuscript has been deemed suitable for publication in PLOS ONE. Congratulations! Your manuscript is now with our production department. 

Kind regards, 

on behalf of

Dr. Alejandro Vega-Muñoz 

Academic Editor

PLOS ONE